# Halogen-Based 17β-HSD1 Inhibitors: Insights from DFT, Docking, and Molecular Dynamics Simulation Studies

**DOI:** 10.3390/molecules27123962

**Published:** 2022-06-20

**Authors:** Arulsamy Kulandaisamy, Murugesan Panneerselvam, Rajadurai Vijay Solomon, Madhavan Jaccob, Jaganathan Ramakrishnan, Kumaradhas Poomani, Muralikannan Maruthamuthu, Nagendran Tharmalingam

**Affiliations:** 1Department of Biotechnology, Bhupat and Jyoti Mehta School of Biosciences, Indian Institute of Technology Madras, Chennai 600 036, Tamil Nadu, India; 2Department of Chemistry and Computational Chemistry Laboratory, Loyola Institute of Frontier Energy, Loyola College, Chennai 600 034, Tamil Nadu, India; panneerchem130491@gmail.com; 3Department of Chemistry, Madras Christian College (Autonomous), Tambaram East, Chennai 600 045, Tamil Nadu, India; 4Laboratory of BioCrystallography and Computational Molecular Biology, Department of Physics, Periyar University, Salem 636 011, Tamil Nadu, India; rjaganphy@gmail.com (J.R.); kumaradhas@yahoo.com (K.P.); 5Division of Pharmacoengineering and Molecular Pharmaceutics, Eshelman School of Pharmacy, University of North Carolina, Chapel Hill, NC 27599, USA; murali.kbiotech@gmail.com; 6Division of Infectious Diseases, Rhode Island Hospital, Alpert Medical School, Brown University, Providence, RI 02903, USA; micronagu@gmail.com

**Keywords:** 17β-HSD1 inhibitors, estrogens, cancer, halogens, DFT, docking, stability, MD simulations

## Abstract

The high expression of 17β-hydroxysteroid dehydrogenase type 1 (17β-HSD1) mRNA has been found in breast cancer tissues and endometriosis. The current research focuses on preparing a range of organic molecules as 17β-HSD1 inhibitors. Among them, the derivatives of hydroxyphenyl naphthol steroidomimetics are reported as one of the potential groups of inhibitors for treating estrogen-dependent disorders. Looking at the recent trends in drug design, many halogen-based drugs have been approved by the FDA in the last few years. Here, we propose sixteen potential hydroxyphenyl naphthol steroidomimetics-based inhibitors through halogen substitution. Our Frontier Molecular Orbitals (FMO) analysis reveals that the halogen atom significantly lowers the Lowest Unoccupied Molecular Orbital (LUMO) level, and iodine shows an excellent capability to reduce the LUMO in particular. Tri-halogen substitution shows more chemical reactivity via a reduced HOMO–LUMO gap. Furthermore, the computed DFT descriptors highlight the structure–property relationship towards their binding ability to the 17β-HSD1 protein. We analyze the nature of different noncovalent interactions between these molecules and the 17β-HSD1 using molecular docking analysis. The halogen-derived molecules showed binding energy ranging from −10.26 to −11.94 kcal/mol. Furthermore, the molecular dynamics (MD) simulations show that the newly proposed compounds provide good stability with 17β-HSD1. The information obtained from this investigation will advance our knowledge of the 17β-HSD1 inhibitors and offer clues to developing new 17β-HSD1 inhibitors for future applications.

## 1. Introduction

Estrogens are a class of sex steroid hormones essential to the female reproductive system [1]. These estrogens are not only used for sexual reproduction, but also play a vital role in regulating cholesterol production and limiting plaque build-up in the coronary arteries [2,3,4,5]. Furthermore, their participation in maintaining the proper balance between bone build-up and breakdown to preserve bone strength is fascinating [4,5]. Among them, the most potent 17β-estradiol (E2) is popularly known for its action through the trans-activation of estrogen receptors (ERs) or through inducing non-genomic effects via the mitogen-activated protein kinase signaling pathway [6]. The importance of 17β-estradiol (E2) is constantly increasing as it has been involved in the treatment of estrogen-dependent diseases (EDDs) in recent years [6,7,8,9]. The standard strategy is to block the action of estrogen using a selective estrogen-receptor modulator (SERM) or anti-estrogens. Often, researchers tend to use aromatase inhibitors (AIs) or gonadotropin-releasing hormone (GnRH) analogs to inhibit estrogen synthesis [10,11,12]. Estrogen-inhibitor-mediated therapies evolved, and availing advanced therapeutical options with specific side effects made researchers focus on the need to eliminate such stresses [13]. The enzyme 17β-HSD1 catalyzes the last step of estrogen’s biosynthesis and transforms estrone (E1) to E2 [14]. Therefore, barring the last step of estrogen biosynthesis by inhibition using the 17β-HSD1 inhibitor can be an attractive approach. Consequently, it is necessary to find new inhibitors with high activity for the specific and selective treatment of estrogen-dependent diseases.

Hydrogen bonds (HBs) are inevitable in biological systems, yet halogen bonds (XBs) are now becoming very popular and of current interest [15,16,17,18,19,20]. XB’s wide applications in supramolecular chemistry [21,22,23], material science [24], molecular recognition [25,26,27], and also in biological systems [28,29,30,31] make this type of non-covalent interaction attractive. Furthermore, Hays et al. showed shorter Br---O contacts at the Holliday junction [32], which kindle our curiosity to substitute halogens in these inhibitors to tune their inhibition activity. Several halogen-based drugs have been approved to treat human diseases over the years [33,34,35]. In spite of several experimental and computational approaches, a detailed systematic computational investigation is clearly lacking, and explaining the binding modes, binding affinity and their respective interactions between the estrogen-related receptor protein and inhibitor will be highly beneficial to understand the biological phenomenon better [36]. Therefore, in the present study, an attempt has been made to observe the characteristic features of halogen-based inhibitors. Keeping these things in mind, the following objectives are framed: 1. To gain insights into the electronic structure of a halogen-substituted inhibitor; 2. To build a mechanistic model to describe how these inhibitors bind with the 17β-HSD1 protein; 3. To find out how these halogen-substituted inhibitors behave differently to the experimentally synthesized and reported 17β-HSD1 inhibitors; 4. To understand the nature of the interactions between the inhibitors and neighboring residues; and 5. To draw some valuable clues to model new effective inhibitors using halogens in the future.

Thus, in this study, 16 inhibitors with certain halogens, such as fluorine (F), chlorine (Cl), bromine (Br), and iodine (I), are modeled as 17β-HSD1 inhibitors (Figure 1). Quantum chemical calculations of these sixteen inhibitors are performed to characterize the electronic structure of these proposed compounds. Additionally, the biological activities of these compounds are addressed through molecular docking. Furthermore, the stability of these docked complexes is also validated by performing a molecular dynamic simulation study. Thus, the present study highlights the importance of halogen in preparing new inhibitors and how these new halogen-substituted inhibitors interact with proteins. Overall, this investigation warrants the optimization of new hydroxyphenyl naphthol-based inhibitors with potent in vitro and in vivo activities in the future.

## 2. Computational Methodology

### 2.1. Calculation of Quantum Chemical Descriptors

Martin Frotscher et al. recently reported sixteen potential 17β-HSD1 inhibitors with the 1-phenyl-hydroxyphenylnaphthol skeleton [13]. Out of them, the reference (R) molecule (in Figure 1) was identified as the more potent one [13]. Therefore, a series of candidates were based on this molecule (R). The literature shows that halogen-substituted molecules increase the possibility of exhibiting halogen bonding when interacting with proteins and DNAs. In recent years, the role of the halogen-bonding interaction has attracted much attention in the inhibitor- and drug-designing fields [20,27,37]. Inspired by these fascinating aspects of halogen bonding, different halogens (F, Cl, Br, and I) have been used in the present study. Sixteen new inhibitors were derived from the reference molecule by substituting halogens at the R_1_, R_2_, and R_3_ positions of sulfonamide containing the 1-phenyl-hydroxyphenylnaphthol (R) compound, as depicted in Figure 1. In the present work, we performed detailed electronic structure calculations on the sixteen halogen-based inhibitors using the Gaussian 09 suite program (Gaussian Inc, Wallingford, CT, USA) [38]. All the geometries were optimized using Becke with the Lee, Yang, and Parr (B3LYP) gradient-corrected correlation functional. The 6-311+G(d,p) basis set without symmetry constraints was used for searching the stationary points [39,40]. As B3LYP performs well for most organic molecules, the same was used here [41,42,43]. The frequency calculation was performed on the optimized geometries to characterize the minima on the potential energy surface. Frequency calculations revealed that all the molecules were found to have present frequencies and no imaginary frequencies were obtained. Various quantum chemical descriptors were calculated using the following theoretical background: Iczkowski and Margrave defined the chemical potential (*µ*) as a negative of the electronegativity as well as the first derivative of the total energy (*E*) for the number of electrons.
(1)$$\chi =-\mu \;=-{\left(\frac{\partial E}{\partial N}\right)}_{\nu (r)}$$

In general, the chemical hardness (*η*) is the second derivative of energy of an atomic and molecular system with respect to the number of electrons (*N*). Hardness is a quantum chemical descriptor used to measure resistance to changes in the electron distribution of a system.
(2)$$\eta \;=\frac{1}{2}{\left(\frac{\partial^{2}E}{\partial N^{2}}\right)}_{\nu (r)}$$

According to the following operational and approximate definitions of Parr and Pearson, the global hardness (*η*) and softness (*S*) are calculated as follows:(3)$$\eta =\frac{\left(IE-EA\right)}{2}\;S=\frac{1}{\left(IE-EA\right)}$$
where ionization energy (*IE*) and electron affinity (*EA*) are the first vertical ionization energy and electron affinity of the molecule, respectively. The global electrophilicity index (*ω*) is calculated by using the chemical potential and hardness as:(4)$$\omega =\frac{\mu^{2}}{2\eta }$$

Furthermore, the ionization energy (*IE*) and electron affinity (*EA*) are calculated based on the Koopmans’ theorem [29] by using the following relation:(5)$$-\varepsilon_{HOMO}=IE-\varepsilon_{LUMO}=EA\;\chi =-\mu =-{\left(\frac{\partial E}{\partial N}\right)}_{\nu (r)}\;$$

The approximate definition of hardness and chemical potential can be written as follows:(6)$$\eta =\frac{\left(E_{LUMO}-E_{HOMO}\right)}{2}\;\mu =\frac{\left(E_{LUMO}+E_{HOMO}\right)}{2}$$

Further molecular docking is also conducted by using minimized geometries obtained from the quantum chemical calculations.

### 2.2. Prediction of Biological Activities of Compounds Using Molecular Docking Analysis

The molecular docking was performed using the AutoDock 4.2 program (The Scripps Research Institute, La Jolla, CA, USA) to understand the nature of interactions between the inhibitors and receptor [44,45]. The crystal structure of Human 17β-hydroxysteroid dehydrogenase type 1 (17β-HSD1) was retrieved from the protein data bank (PDB code: 1FDT) [46] with a resolution of 2.20 Å (Figure 1). After removing the additional water molecules and all the heteroatoms from the protein, Kollman united atom charges and polar hydrogen atoms were added to the protein in the docking simulation. The DFT optimized geometries were used directly for docking with charged proteins. The AutoGrid program was used to produce the grid maps that covered the active site of proteins with dimensions of 40 × 40 × 40 Å, and the spacing between each grid point was around 0.375 angstroms. The grid center was set at 45.41, 5.71, and 40.85 for x, y, and z. A semi-flexible docking approach using a genetic algorithm–least squares (GA–LS) technique was performed with the AutoDock 4.2 program. Finally, the lowest energy with top-ranked protein–ligand conformation was chosen. Furthermore, the noncovalent interactions, such as hydrogen and π–interactions between 17β-HSD1 and compounds, were visualized by Discovery studio [47]. Additionally, the halogen-bond interactions were computed based on a distance of <4 Å between hydrogen atoms of 17β-HSD1 and halogens using in-house Perl scripts. The halogen-bond interactions were visualized in the PyMOL molecular graphics system [48].

### 2.3. Assessment of the Stability of Docked Complexes Using Molecular Dynamics (MD) Simulations

We conducted molecular dynamics simulation studies to explore the stability of halogen-substituted inhibitors binding to a 17β-HSD1 receptor. The top four docked complexes were selected for this MD simulation based on docking binding-free energy/score and intermolecular interactions. We performed the 100 ns molecular dynamics simulations using by Schrödinger DESMOND MD package (Schrödinger, New York, NY, USA) with an OPLS4 force field [49,50,51]. Furthermore, the binding stability and conformational modifications of selected complexes were examined by analyzing the trajectories in terms of RMSD (root-mean-square deviation) and RMSF (root-mean-square fluctuation).

## 3. Results and Discussion

### 3.1. Structural Features of Halogen-Substituted Compounds against 17β-hydroxysteroid Dehydrogenase Type 1

New inhibitors were prepared by substituting halogens (F, Cl, Br, and I) at various positions of the potent compound reported by Rolf W. Hartmann et al. (Figure 1). We chose R_1_, R_2_, and R_3_ positions as the positions suitable for chemical modification and they were easy to substitute with different functional groups. Moreover, the ease of synthesizing these compounds is essential in designing the molecules and it provides flexibility to the researchers to tune the structures. Therefore, the OH group in R_2_ and R_3_ and hydrogen (H) in R_1_ were replaced by halogen atoms. B3LYP optimized geometries are given in Figure 1. Our results show that all the molecules have excellent π-delocalization throughout the molecule. For instance, the C–C bonds connecting the naphthyl ring with benzene rings (the bridging C–C bond lengths) were found to be ~1.48 Å (d2) to ~1.49 Å (d1), which lie between their single- and double-bond limits. The dihedral angle (Φ1,2) between the naphthyl ring and the adjacent benzene rings was affected mainly due to halogen substitution (−35 to −90°), shown in Figure 1. Due to the larger size of the iodine atom, the dihedral angle was distorted to a greater extent (~90° for Φ1) than other halogens (Figure 1).

Interestingly, from the optimized geometries, we observed that the substitution at R_1_ opened the possibility of showing weak interactions with a neighboring hydrogen atom. This plays a vital role in decreasing the total energy and stabilizing the molecule. Thus, our results imply that halogen substitution provides stabilization to these molecules.

### 3.2. Characterization of Halogen-Derived Compounds Using Quantum Chemical Descriptors

It is essential to correlate the quantum chemical descriptors with the inhibitors’ biological activity that provides direct information about the reactivity of molecules. Therefore, the present work aimed to emphasize how far quantum chemical descriptors help to correlate the biological reactivity of these halogen-substituted hydroxyphenyl naphthol derivatives. Appendix A lists various quantum chemical descriptors computed using the optimized geometries of halogen-substituted inhibitors. Concerning the experimentally reported highly potent molecule (Reference (R) in Appendix A), halogen substitution in the three different positions of the hydroxyphenyl naphthol significantly affected the quantum chemical descriptors. This will be discussed in the following sections.

#### 3.2.1. Frontier Molecular Orbital Analysis

The frontier molecular orbitals (FMOs) are often used to derive qualitative information about the electronic structure properties of molecules and estimate the chemical reactivity of the molecules. The frontier molecular orbitals (FMOs) of the sixteen compounds starting from HOMO−1 to LUMO+1 levels, along with the FMO gap of the inhibitors, are presented in Figure 2. The figure clearly shows that the HOMO–LUMO gap of halogen-substituted molecules is less than that of R in _Cl_R_1_, _Br_R_1_, and _I_R_1_ positions. However, the substitution of fluorine at R_1_, R_2_, and R_3_ positions does not strongly influence the energy gap despite their HOMO and LUMO levels, which were slightly perturbed. In all the cases, the LUMO energy levels were reduced as we increased the number of halogen substitutions. This was predominant when the halogen atom with a higher atomic number was heavier. Notably, the energy gaps of _I_R_1_ and _I_R_4_ were found to be 3.08 eV and 3.45 eV, respectively. This implies that the substitution of halogens in the R_1_ position significantly reduced the LUMO levels and increased the electrophilicity of all 16 molecules. Furthermore, _I_R_1_ and _I_R_4_ molecules showed greater electrophilicity values of 5.36 and 5.83, respectively, compared to all other molecules.

Furthermore, the substitution of iodine in the R_1_ and R_4_ positions of the reference molecule affected the LUMO energy levels of _I_R_1_ and _I_R_4_ predominately (1.16 eV and 0.78 eV). This clearly shows that a heavy iodine atom was found to stabilize the LUMO levels, increasing the electrophilic nature of these halogen-substituted molecules.

In the present scenario, the halogen atom acts as an electron-acceptor site, thereby increasing the tendency of the molecules to interact with the electron donors through the halogen-bonding interaction. The halogen-bonding interaction was found to increase with the increasing polarizing nature of the halogens and an increase in the atomic number of halogen atoms. Halogen-bonding interactions are presumed to be predominant in the case of iodine-substituted compounds _I_R_1_ and _I_R_4_. The ground state density plots of HOMO and LUMO levels of sixteen molecules and reference compounds are shown in Figure 3 (values are presented in Appendix A Appendix A).

From the HOMO–LUMO plot of the reference compound, one can understand that the HOMO was mainly localized over the phenyl-naphthol unit, while the LUMO originated in the naphthol units. Looking at the HOMOs, it is clear that all these molecules have a very similar distribution, while their LUMOs are significantly affected. For instance, the LUMOs in _Cl_R_1_, _Cl_R4, _Br_R_1_, _Br_R_4_, and _I_R_1_ and _I_R_4_ were different when substituting halogens (Cl, Br, and I) in the nitrogen atom of the sulfonamide. In these cases, the whole electronic distribution of LUMO is concentrated only in the sulfonamide group (most preferably in the N–X bond [X = Cl, Br, and I]).

The contribution of HOMO in _Cl_R_1_, _Cl_R_4_, _Br_R_1_, _Br_R_4_ and _I_R_1_ and _I_R_4_ was primarily concentrated in the hydroxyphenyl naphthol group. In contrast, the LUMO level was largely populated in the N–I bond of phenyl sulfonamide groups. In _Cl_R_1_, _Cl_R_4_, _Br_R_1_, _Br_R_4_, and _I_R_1_ and _I_R_4_, the LUMO level was populated in the phenyl sulfonamide group and not in the hydroxyphenyl naphthol group. Furthermore, iodine substitution (in R_1_ and R_4_) enhanced the reactivity of the inhibitor molecules through the reduced H–L gap more than the other halogen counterparts. Therefore, the electrophilicity of the iodine-substituted inhibitors substantially increased by lowering their LUMO energy values. Significantly, the reduction in LUMO energy levels was highest in _I_R_1_ and _I_R_4_. This clearly shows that the substitution of iodine increases the reactivity of the inhibitors, preferably at R_1_ and R_4_ positions. This kind of striking difference in the electronic density population and contributions of FMO levels considerably increase the inhibition activity of these molecules. More specifically, this effect is expected to be predominant in the case of iodine substituted at R_1_ and R_4_ positions.

#### 3.2.2. Hardness and Softness

Hardness (*η* in eV) and softness (*S* in eV) parameters are used to study an indicative index of the stability of chemical molecules. Softness has an inverse relationship with the hardness of the molecule. We identified that the hardness value of a reference compound was 2.12 eV, and, for the sixteen molecules, it fell in the range of 1.54 to 2.12 eV (Appendix A). More specifically, this effect was much more pronounced when substituting halogens in the nitrogen atom of the hydroxyphenyl naphthol molecule (R_1_ position), i.e., _Cl_R_1_, _Br_R_1_, and _I_R_1_ (1.99, 1.74, and 1.54, respectively). The minimum hardness value of _I_R_1_ indicates that the substitution (R_1_ position) of halogens facilitates a higher reactivity pattern, leading to a lower hardness value than the other molecules. Similarly, molecules _Cl_R_1_, _Br_R_1_, and _I_R_1_ were found to have slightly lower softness values than all the other compounds. This indicated that the R_1_-substituted halogen molecules may be expected to have a similar reactivity pattern and higher binding affinity with estrogen receptors.

#### 3.2.3. Electrophilicity (*ω*) and Chemical Potential (*µ*)

The electrophilicity index is an essential parameter for depicting the reactivity of a particular molecule. This quantum chemical descriptor can be used to measure the energy stabilization of a molecule when it acquires additional electronic charge from its surroundings. Another critical global reactivity descriptor is chemical potential (*µ*), which measures the escaping tendency of an electron. The higher reactivity of the molecules is always attributed to large negative µ and high positive *ω* values. From Appendix A, it can be observed that the selected sixteen molecules were found to have large negative *µ* and high positive *ω* values, which is highly indicative that these molecules may have a higher interaction affinity and may be highly potent drug molecules compared to our reference molecules.

### 3.3. Molecular Electrostatic Potential (MESP) Analysis

Over the years, the local relativities of the molecules can be calculated to analyze the molecular electrostatic potential analysis (MESP), which is a powerful tool to identify the electrostatic interactions of the electron-rich and electron-deficient regions of the molecules.

Additionally, it helps to predict the possible site of interactions of drug molecules when they bind in the active site of a protein. Therefore, the electrostatic potential map of the 16 molecules was generated and presented in Figure 4. These MESP maps allows us to visualize the variably charged regions of an electron-rich and -deficient molecule. Here, the blue region indicates the positively charged (non-reactive) sites, the red region suggests the more electron-rich negatively charged site (i.e., electrophilic attack site), and the green region indicates the overall zero-potential locations of the molecules. Among all the molecules, the phenyl sulfonamide groups showed the red region of the negatively charged site of the molecules.

Overall, our DFT calculations expand our knowledge of their structure and electronic properties. More specifically, the analysis of FMOs clearly illustrates that the LUMO level upon the substitution of heavier halogen atoms is lower. This trend is observed in all the classes of molecules we studied in this work. This effect is much more pronounced, especially in the iodine-substituted (in R_1_ and R_4_ positions) hydroxylphenyl naphthol derivatives. A highly electrophilic iodine atom increases the electrophilic character of an entire molecule and significantly changes the electronic density population. The tri-substitution of halogens had a considerable impact on their molecular properties, compared to single and double substitutions. Various descriptors calculated in the present study indicated that halogen significantly alters these molecules’ molecular properties and chemical reactivity. It is interesting to study how these DFT descriptors were connected to their biological activity. Hence, detailed molecular docking analysis followed by molecular dynamics simulations were performed, and the results are discussed here.

### 3.4. Predicting the Biological Activities of Halogen-Substituted Ligands against 17β-HSD1 Validation of Molecular Docking Using a Redocking Approach

We utilized the molecular docking strategy for identifying the possible binding orientations and conformations of 1-substituted hydroxyphenyl-2-naphthols derivatives in the binding pocket of 17β-HSD1. To validate the reliability of the docking procedure, we used the redocking methodology [52,53] that removed the native ligand (EST) from the 17β-HSD1 complex (1FDT) and docked it again in the same pocket/orientation using AutoDock. Next, we calculated the root-mean-square deviation (RMSD) of 1.03 Å between the original complex and our redocked complex.

This observation suggests that the molecular docking procedure is robust and the binding orientation is similar to a native complex (Figure 5a). Furthermore, the redocking parameters were used to conduct all the new compounds (**1**–**16**), and we observed that all the molecules had a similar binding mode. The binding orientation of all the halogen-substituted compounds in the 17β-HSD1 receptor are shown in Figure 5b.

#### Analysis of Noncovalent Interactions between 17β-HSD1 and Halogen-Substituted Ligands

Interestingly, the molecular docking results show that the 16 halogen-substituted compounds achieved the highest binding-free energy with −10.25 to −11.94 kcal/mol in all the cases, compared to the reference molecule (−10.21 kcal/mol). Among the different types of halogens (F, Cl, Br, I), we observed that the overall binding-free energy trend was F < Cl < Br < I. In addition, for the binding-free energy comparison within each halogen subgroup, the maximum values were obtained for _F_R_4_, _Cl_R_4_, _Br_R_4_, and _I_R_4_ compounds with halogen atoms at all three positions: R1, R2, and R3 (Figure 1). Notably, the DFT studies indicated that these four halogen compounds had a large, positive electrophilicity (*ω*) and a significant negative value of chemical potential (*µ*) than the others. Additionally, the substituted halogens might provide a stronger binding affinity towards the 17β-HSD1 through halogen-bonding interactions. The docking binding-free energies/scores and inhibition constant; the total number of bonds in different types of noncovalent interactions, such as hydrogen, π-bonds, and halogen bonds; hydrogen; π; and halogen-bond interacting residues are summarized in Appendix A. The molecular docking results indicate that the halogen-derived compounds might act as potential 17β-HSD1 inhibitors in breast cancer.

Furthermore, we analyzed the contribution of different noncovalent bond interactions in the docked complexes of 17β-HSD1 and 16 halogen-derived compounds (Figure 6 and Figure 7). We identified four hydrogen bonds, eight π interactions, and there were no possibilities for halogen-bond interactions in the reference compound; however, our halogen-substituted compounds had a greater number of above-mentioned different types of interactions than the reference molecule. Among the fluorine-substituted compounds, the potent compound _F_R_4_ (high free energy of −10.53 kcal/mol) showed 3 hydrogen-bond interactions and 11 π interactions with Gly186, Val188, His221, and Val143, Met147, Leu149, Tyr155, Cys185, Pro187, Val225, and Phe226 residues, respectively. Interestingly, this compound had a high tendency to bind with a hydrophobic pocket of 17β-HSD1 residues by making 22 halogen-bond interactions. This revealed that the binding pockets were mainly enriched with hydrophobic residues, which are essential for stabilizing this complex (Figure 6b and Figure 7a).

Recent studies showed that the protein–protein, protein–small molecule complexes were mainly stabilized by hydrophobic residues, such as Leu, Val, and Phe [54,55]. Additionally, 18 hydrogens, halogen, and π interactions were observed in _cl_R_4_, which had a binding score of −11.58 kcal/mol and an affinity of 3.27 nM (Figure 6c and Figure 7b). The _Br_R_4_ had ~30 noncovalent interactions with the inhibitory constant of 2.59 nM, and the binding pocket contained the combination of different physicochemical properties of residues (for example, Val143, Tyr155, Pro187, His221). The maximum binding-free energy was observed in compound (iodine-substituted; −11.94 kcal/mol; K_i_ = 1.78 nM) _I_R_4_ and it formed 13 halogen-bonding interactions and 9 hydrogen–π interactions (Figure 6e and Figure 7d). The high-binding-interaction energy and low K_i_ value of compound IR4 imply that the presence of three iodine atoms facilitated the enhancement of the electrophilicity of the entire molecule through the lowering of the energy of the LUMO level, leading to high electrophilicity values. This enhanced the possibility of forming a particular class of weak interactions called halogen bonding. This interaction is vital in stabilizing the drug molecules in the protein pocket and enhancing the inhibition activity.

Molecular docking studies identified the following key observations: (i) all the halogen-substituted compounds showed a robust binding affinity against 17β-HSD1; (ii) a significant number of halogen interactions were observed in these compounds; (iii) specifically, the substitution of halogen atoms in the R_1_, R_2_ and R_3_ positions of reference compound substantially increased the inhibition activity and higher binding affinity towards 17β-HSD1; (iv) iodine-derived compounds were more potent than other halogens, and the binding energy trend was followed in the manner of F < Cl < Br < I; (v) the binding pocket of all the halogen-substituted compounds mostly occupied 90% of hydrophobic residues; and (vi) the residues, such as Val143, Val149, Pro187, Val188, and His221, were the residues with the most potential to interact with ligands.

### 3.5. Relationship between DFT Descriptors and Predicted Biological Activities (Binding Energy/Score) of Halogen-Substituted Compounds

To characterize the descriptors and to understand the properties of these molecules with their biological activities, we computed the Pearson correlation between the binding-free energy/docking score and DFT descriptors, such as softness (*S*), hardness (*η*), electrophilicity (*ω*), chemical potential (*µ*), and energy gap (*Eg*).

Interestingly, we identified that the chemical potential and electrophilicity have a high positive (r = 0.81) and negative (−0.77) correlations with the binding energy/score correlation. For the rest of the properties, we obtained a positive correlation of around 0.6 (Figure 8). We observed that the highest binding-affinity ligand must have a greater electronegativity than the other ligands (Figure 8a) and a lower chemical energy, softness, hardness, energy gap than the others (Figure 8b–d). These characterizations of 17β-HSD1 inhibitors revealed the relationship between the molecule activity, and were ultimately used to improve the biological activity of these molecules.

### 3.6. Investigation of the Structural Stability of Halogen-Substituted-Inhibitor Complexes from MD Studies

We conducted the RMSD and RMSF parametric analyses of the molecular dynamic simulation of docked complexes to understand the standard displacement of the atoms and internal fluctuations of each amino acid residue in the complex during the simulation, respectively. We found that the _F_R_4_, _Cl_R_4_, _Br_R_4_, _I_R_4_, and reference complexes showed the RMSD deviations in the ranges of 1.6–3.6, 1.5–3.5, 1.5–2.4, 1.3–3.2, and 1.5–2.3 Å, respectively (Figure 9a). Based on the halogen-atom types, 17β-HSD1 with bromine showed more excellent structural stability, followed by iodine, fluorine, and chlorine. This observation confirms the strength of each complex during the 100 ns molecular dynamics simulations. On the other hand, the amino acid residues’ fluctuations in the range are nearly the same for all the complexes due to the same protein present in the complexes (Figure 9b).

Specifically, the binding residue regions, such as 140–160, 180–200, and 80–90, showed fewer fluctuations and stable interactions throughout the simulations. These results indicate that all the complexes show the minimum deviation during the MD simulations, confirming the structural stability of the complexes.

## 4. Conclusions

In the present work, an attempt was made to prepare halogen substituted in 17β-HSD1 inhibitors and evaluate their electronic structure and biological properties through a detailed DFT, molecular docking, and molecular dynamics simulations. The results reveal that halogen substitution enhances the efficiency of the reference molecule. The DFT calculations clearly show that halogens play a vital role in stabilizing the LUMO more than the HOMO level. Halogen substitutions at three different positions were found to be effective by reducing the HOMO–LUMO energy gap. Molecular docking analysis helps us to understand the nature of the interactions between the inhibitors and the protein, and to rank these halogen-substituted inhibitors according to their inhibition efficiency. Halogen substitution increases the chance of halogen interactions inside the protein environment in addition to the hydrogen-bonding possibilities, which enhances the inhibition activity of molecules. Furthermore, the 100 ns molecular dynamics study indicates that these inhibitors are very comfortable inside the active site of the protein and supports the claim for using them as possible inhibitors. In summary, this work sheds light on the importance of halogens in producing new efficient inhibitors to treat estrogen-dependent diseases and encourages researchers to tune the structures of drug candidates using halogens.

## Data Availability

Data are contained within the article and Appendix A.

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
