# Peer review of "Halogen-Based 17β-HSD1 Inhibitors: Insights from DFT, Docking, and Molecular Dynamics Simulation Studies"

_molecules, 2022, doi:10.3390/molecules27123962_

Round 1

Reviewer 1 Report

The Authors keep using word „design”, while In fact they have just analyzed those compounds in silico. I would suggest using different word.

In the introduction the Authors should present some more information on the application of molecular modelling methods in the studies of estrogens and estrogen-related enzymes to provide better background. Besides, it should be stated why calculating the HOMO LUMO is so important in this particular case?

Figure 1, have the Authors performed any conformational search for the global minimum?

Figure S1 should be put into the manuscript from SI.

Line 263, not “highest” but “higher”

Line 334, this should be somehow discussed. For example, in order to posses the highest affinity the “ideal” ligand should have electronegativity higher than… and chemical potential lower than… .

2.6. In this part the Authors should state why they have used both RMSD and RMSF and what the observed values actually mean.

Lines 362-370, this should not be a part of “Materials and methods”. I recommend removing this part.

Have the Authors used any solvation scheme (SMD, PCM)?

Have the Authors used any dispersion corrections (i.e. D3)?

I just wonder, why the Authors used Autodock for docking while they had an access to the Schrodinger suite? The molecular docking using Schrodinger is more accurate since it allows induced fit docking and subsequent MM-GBSA calculations.

Lines 437-439, so the Authors have performed 4 MD runs for each ligand? Am I correct? If yes, where are those results?

Author Response

We thank the reviewers for their constructive comments. We have carefully addressed all the points and the corrections are incorporated in the revised manuscript.

REVIEWER 1:

  • The Authors keep using word “design”, while In fact they have just analyzed those compounds in silico. I would suggest using different word.

Authors’ Response: We wish to bring to your kind notice that in this work, we have taken a known compound and their derivatives have been prepared through halogen substitutions. These prepared molecules are new and not yet known. Therefore, we prefer the word ‘design’ to other words. We believe that the reviewer will graciously accept our choice.   

  • In the introduction the Authors should present some more information on the application of molecular modelling methods in the studies of estrogens and estrogen-related enzymes to provide better background. Besides, it should be stated why calculating the HOMO LUMO is so important in this particular case?

Authors’ Response: We accept the reviewer’s suggestion of including the importance of molecular modeling and therefore the same is included in the revised manuscript. When it comes to HOMO-LUMO calculations, here we have looked into the electronic structure of these molecules through DFT calculations and the calculation of HOMO and LUMO energy values are very vital in estimating not only the chemical reactivity of molecules but also the kinetic stability. This has a huge application in predicting the ability of the molecules to interact with other biological entities through intramolecular charge transfer which is used in Pharmaceutical studies [REF1].

REF1: Mabkhot YN, Aldawsari FD, Al-Showiman SS, Barakat A, Soliman SM, Choudhary MI, et al. (2015) Novel enaminone derived from thieno[2, 3-b]thiene: Synthesis, x-ray crystal structure, HOMO, LUMO, NBO analyses and biological activity. Chem. Cent. J 9: 1–11.

  • Figure 1, have the Authors performed any conformational search for the global minimum?

Authors’ Response: Yes. We performed the conformational analysis to identify the most stable structure of the parent molecule. In addition to that, we wish to bring to your kind notice that optimization is followed by frequency calculations to ensure the global minima on the potential energy surface by obtaining all real frequencies for all the molecules. This has been now clearly mentioned in the materials and methods section 2.1 of the revised manuscript (Page 3, Lines 111-114). 

  • Figure S1 should be put into the manuscript from SI.

Authors’ Response: We moved the figure into main manuscript and changed the figure number accordingly.

  • Line 263, not “highest” but “higher”

Authors’ Response: The suggestion is included in the revised version of manuscript.

  • Line 334, this should be somehow discussed. For example, in order to posses the highest affinity the “ideal” ligand should have electronegativity higher than… and chemical potential lower than… .

Authors’ Response: Thanks for your constructive comment. We discussed the results in the following way in the revised version of manuscript: We observed that the highest binding affinity ligand must have a greater electronegativity than other ligands (Figure 8) and lower chemical energy, softness, hardness, energy gap than others. (Page 13, lines: 443-447).

  • 2.6. In this part the Authors should state why they have used both RMSD and RMSF and what the observed values actually mean.

Authors’ Response: RMSD is widely used to determine the macromolecular structures similarities of the trajectory frames which could be obtained during the MD simulations with target structure as reference. It is used to know the large changes in the protein structure as compared to the initial structure. RMSF is used to understand the fluctuation around an average per residue of the macromolecular structure during the MD simulation. It calculates individual amino acid residue flexibility of the target protein and the residues fluctuations over the MD simulation (Page 4, lines:177-183).

  • Lines 362-370, this should not be a part of “Materials and methods”. I recommend removing this part.

Authors’ Response: We have changed the title of this part of “Computational Methodology” and the section number is also changed as 2.1.

  • Have the Authors used any solvation scheme (SMD, PCM)?

Authors’ Response: No, all the calculations have been done in gas phase and is now mentioned in the revised manuscript. (section 2.1)

  • Have the Authors used any dispersion corrections (i.e. D3)?

Authors’ Response: We wish to bring to your kind notice that we did not include dispersion corrections as we calculate the electronic properties of individual molecules. 

  • I just wonder, why the Authors used Autodock for docking while they had an access to the Schrodinger suite? The molecular docking using Schrodinger is more accurate since it allows induced fit docking and subsequent MM-GBSA calculations.

Authors’ Response: We thank the reviewer for the question and we wish to bring to your kind notice that we don’t have access to Schrodinger suite. Therefore, we used Autodock for the docking calculations because it is an open-source software and the results can be reproducible by a common man.

  • Lines 437-439, so the Authors have performed 4 MD runs for each ligand? Am I correct? If yes, where are those results?

Authors’ Response: We performed the MD simulation to understand the stability of the complexes by analyze the trajectories and plotting the RMSD and RMSF.  In 3.6, both RMSD and RMSF values are presented for all four complexes by analyzing corresponding trajectories which are obtained from the MD simulation studies of each complex. Also, the results are presented in Figure 9.

Reviewer 2 Report

This article presents computational results on studying the binding of halogen derivatives of a parent compound to 17β-hydroxysteroid dehydrogenase type 1. The authors reported several properties of the compounds computed by quantum chemistry, binding energy and conformation obtained by molecular docking, and the stability of the docking structures by molecular dynamics. In general, the calculations appeared to show that halogen substitutions could improve binding to the protein, and the binding affinity correlated with the chemical potential and the electrophilicity computed by quantum mechanics.

The authors might want to clarify how lower energy gap gave rise to excellent stability, and what stability did they refer to.

Needs to provide more explanation on how the frontier molecular orbitals are related to the binding affinity to the protein. Likewise for hardness, softness, and reactivity. In its present form, the concept is vague.

Some typos need to be corrected and the English needs to be improved, especially to remove some unclear sentences.

Some parts of the paper was not written in a logical manner. For example, the first paragraph of the introduction contained a list of loosely connected thoughts rather than a coherent argument for the proposed study. The conclusion was not logically written to show the usefulness of the work.

Some words appear missing in the sentence: “This implies that substitution of halogens in the R1 position, thereby increasing, i.e., on the nitrogen atom, significantly reducing LUMO levels energy, thereby increasing the electrophilicity of all newly designed 16 molecules.”

The figure captions need more detail. For example, useful to define d1, d2, phi1, and phi2 in Figure 1. Figure 4 needs explanation on what are the three surfaces displayed for each molecule. In addition, the resolution of Figure 5 needs to be improved.

Author Response

We thank the reviewers for their constructive comments. We have carefully addressed all the points and the corrections are incorporated in the revised manuscript.

REVIEWER 2:

This article presents computational results on studying the binding of halogen derivatives of a parent compound to 17β-hydroxysteroid dehydrogenase type 1. The authors reported several properties of the compounds computed by quantum chemistry, binding energy and conformation obtained by molecular docking, and the stability of the docking structures by molecular dynamics. In general, the calculations appeared to show that halogen substitutions could improve binding to the protein, and the binding affinity correlated with the chemical potential and the electrophilicity computed by quantum mechanics.

  • The authors might want to clarify how lower energy gap gave rise to excellent stability, and what stability did they refer to.

Authors’ Response: We thank the reviewer for this question and we wish to inform you that the lower HOMO-LUMO gap always associated with the chemical reactivity and kinetic stability. This has been clearly mentioned now in section 2.1 (Page Number:5, Lines: 220-222).

  • Needs to provide more explanation on how the frontier molecular orbitals are related to the binding affinity to the protein. Likewise for hardness, softness, and reactivity. In its present form, the concept is vague.

Authors’ Response: It is well known that Conceptual Density Functional Theory that provides useful insights in understanding or interpretation/prediction of experimental/theoretical reactivity [REF 1-4] data based on a series of response functions to perturbations in the number of electrons and/or external potential.[REF 5-7] Thus the properties predicted through DFT calculations help us to understand the binding ability of the molecules to proteins to certain extent. Some references given below for the same.

  1. Belaidi S, Melkemi N, Conformational analysis and physical-chemistry property relationship for 22-membered macrolides, Asian J Chem 25(8):4527–4531, 2013
  2. Belaidi S, Mazri R, Mellaoui M, Kerassa A, Belaidi H, Electronic structure and effect of methyl substitution in oxazole and thiazole by quantum chemical calculations, J Pharm Biol Chem Sci 5(3):811–818, 2014.
  3. Almi Z, Belaidi S, Lanez T, Tchouar N, Structure activity relationships, QSAR modeling and drug-like calculations of TP inhibition of 1,3,4-oxadiazoline-2-thione derivatives, Int Lett Chem Phys Astron 18:113–122, 2014.
  4. Mazri R, Belaidi S, Kerassa A, Lancz T, Conformational analysis, substituent e®ect and structure activity relationships of 16-membered macrodiolides, Int Lett Chem Phys Astron 14(2):146–167, 2014.
  5. Geerlings P, De Proft F, Conceptual DFT: The chemical relevance of higher response functions, Phys Chem Chem Phys 10(21):3028–3042, 2008.
  6. Melkemi N, Belaidi S, Salah T, Daoud I, A DFT-based QSARs of some 1,2-Dithiole-3- thione derivatives as inducers of quinine reductase, Res J Pharm Biol Chem Sci 6(3):2017–2024, 2015
  7. Manoj KH, Arup B, Many-electron problem in terms of the density: From Thomas–Fermi to modern density-functional theory, J Theor Comput Chem 02(2):301–322, 2003.

  • Some typos need to be corrected and the English needs to be improved, especially to remove some unclear sentences.

Authors’ Response: We thank the reviewer profoundly for taking time to patiently review the typos and languages issues as well. Throughout the manuscript typos were corrected and the language has been proofread and improved in the revised version.

  • Some parts of the paper was not written in a logical manner. For example, the first paragraph of the introduction contained a list of loosely connected thoughts rather than a coherent argument for the proposed study. The conclusion was not logically written to show the usefulness of the work.

Authors’ Response: Taking into consideration, the reviewer’s comments, we have perused the entire manuscript and we now believe that we have provided a logical discussion in the introduction and conclusion and have properly highlighted the usefulness of the work as well in the revised manuscript. 

  • Some words appear missing in the sentence: “This implies that substitution of halogens in the R1 position, thereby increasing, i.e., on the nitrogen atom, significantly reducing LUMO levels energy, thereby increasing the electrophilicity of all newly designed 16 molecules.”

Authors’ Response: We agree with the reviewer’s comment. This sentence has been corrected in the revised manuscript which has been highlighted in the page no.4 (Page 6, lines:230-232).

  • The figure captions need more detail. For example, useful to define d1, d2, phi1, and phi2 in Figure 1. Figure 4 needs explanation on what are the three surfaces displayed for each molecule. In addition, the resolution of Figure 5 needs to be improved.

Authors’ Response: We have given clear explanation and the full form of the d1, d2, Φ1 and Φ2 in the revised manuscript which is highlighted in footnote of Figure 1(Page 6: Figure1). In Figure 4, We have clearly mentioned on the surface differences in revised manuscript (Page Number 9; Figure 4, line:321). Further, we also improved the quality of figure 6 in the revised version (Page 10: Figure 6).

Round 2

Reviewer 1 Report

I am not satisfied with the way the Authors have answered to my comments and corrected their maniuscript.

First of all, the role of HOMO-LUMO calculations in the design of new ligands should be explained. Personally, I am not convinced why the Authors have calculated those parameteres. This is also mentioned by the second Reviewer.

Secondly, "design" is a catchy word but it doesn't mean simple drawing new structures and checking their affinity in silico.

Further, the Authors claim that they didn't have access to the Schrodinger software. However, in line 173 they state that "Schrödinger DESMOND MD package" (has been used). So what is the true?

The English grammar and style must be greatly improved.

Last, but not least, the Authors claim that they have performed some search over the conformational space of the ligands. However, it is not mentioned in the manuscript. This must be described in details. Have you performed a systematic search? Or a Boltzman jump method? How you have done this. Simple IR frequencies calculations does not ensure that what the Authors have found is not a local, shallow minimum.

Author Response

Reply to reviewers' comments

Reviewer 1.

Comment: I am not satisfied with the way the Authors have answered to my comments and corrected their maniuscript. First of all, the role of HOMO-LUMO calculations in the design of new ligands should be explained. Personally, I am not convinced why the Authors have calculated those parameteres. This is also mentioned by the second Reviewer.

Response: We are sorry to hear about the dissatisfaction delivered. We are trying our best to serve the purpose, and at-most intense is to address the Reviewer's comment with only 100% science involved. We wish to bring to the Reviewer’s kind notice that the HOMO-LUMO calculations give insights into the chemical reactivity and kinetic stability of the molecule, it is common in literature [REF 1-5], and therefore, the same has been adopted here. We have also made a correlation analysis between the DFT parameters and the biological activities. However, as per the Reviewer's comment, this portion is reduced in the revised manuscript.

REF 1: VineetKumar Choudhary, ArvindKumar Bhatt, Dibyajit Dash, Neeraj Sharma, J. Comput. Chem 2019, 40, 2354—2363.

REF 2: Murugesan Panneerselvam, Arunkumar Kathiravan, Rajadurai Vijay Solomon and Madhavan Jaccob Phys. Chem. Chem. Phys., 2017, 19, 6153—6163.

REF 3: Murugesan Panneesrselvam, Madhu Deepan Kumar, Madhavan Jaccob, and Rajadurai Vijay Solomon, ChemistrySelect 2018, 3, 1321—1334.

REF 4: R. V. Solomon, P. Veerapandian, S. A. Vedha, P. Venuvanalingam, J. Phys. Chem. A 2012, 116, 4667—4677.

REF 5: Rajadurai Vijay Solomon, Antony Paulraj Bella, Swaminathan Angeline Vedha and Ponnambalam Venuvanalingam Phys. Chem. Chem. Phys., 2012, 14, 14229—14237.  

Comment: Secondly, "design" is a catchy word but it doesn't mean simple drawing new structures and checking their affinity in silico.

Response: We agree with the Reviewer to modify the term. The word "design" is now replaced with "halogen-based" throughout the manuscript.

Comment: Further, the Authors claim that they didn't have access to the Schrodinger software. However, in line 173 they state that "Schrödinger DESMOND MD package" (has been used). So what is the true?

Response: We would like to mention that we do have a Schrödinger DESMOND MD package. However, we don't hold the Glide - Schrödinger molecular docking package license.

Comment: The English grammar and style must be greatly improved.

Response: The manuscript is thoroughly checked for grammatical errors and word issues. We believe the language of this version is substantially improved, and we have highlighted the changes in the manuscript.

Comment: Last, but not least, the Authors claim that they have performed some search over the conformational space of the ligands. However, it is not mentioned in the manuscript. This must be described in details. Have you performed a systematic search? Or a Boltzman jump method? How you have done this. Simple IR frequencies calculations does not ensure that what the Authors have found is not a local, shallow minimum.

Response: We would like to mention that we have performed molecular mechanics (MM) optimization with various possible orientations to ascertain the most suitable geometry, which is further considered for DFT optimization. Also, we would like to mention that the energy minimization process uses the Berny algorithm implemented in Gaussian09 software. The calculation of harmonic frequencies confirms the ground state geometry by showing all real frequencies.

Reviewer 2.

Comment: The manuscript has been slightly improved. Perhaps can be published.

Response: We would like to thank Reviewer 2 and accepting our thoughts.

Reviewer 2 Report

The manuscript has been slightly improved.  Perhaps can be published.

Author Response

(The authors gave the same response as above.)
